# A Numerical Parametric Study of a Double-Pipe LHTES Unit with PCM Encapsulated in the Annular Space

Evdoxia Paroutoglou [1,*], Peter Fojan [2], Leonid Gurevich [2], Simon Furbo [3], Jianhua Fan [3], Marc Medrano [4] and Alireza Afshari [1]

1   Department of Energy Performance, Indoor Environment and Sustainability of Buildings, Aalborg University, 2450 København, Denmark
2   Department of Materials and Production, Aalborg University, 9220 Aalborg, Denmark
3   Department of Civil and Mechanical Engineering, Technical University of Denmark, 2800 Kongens Lyngby, Denmark
4   Department of Computing and Industrial Engineering, University of Lleida, 1300 Lleida, Spain
*   Correspondence: evp@build.aau.dk

**Abstract:** Latent heat thermal energy storage (LHTES) with Phase Change Materials (PCM) represents an interesting option for Thermal Energy Storage (TES) applications in a wide temperature range. A tubular encapsulation model of an LHTES with PCM was developed, and the calculated data were analyzed. In addition, a parametric analysis for the preferable system geometry is presented. Organic paraffin RT18 with a melting point of 18 °C was utilized as PCM for different geometries of LHTES, and the addition of internal and external fins and their influence on LHTES thermal conductivity was investigated. One-step heat exchange from outdoor air to PCM and from PCM to water characterizes the LHTES system in solidification and melting processes, respectively. A 2D axisymmetric model was developed using Comsol Multiphysics 6.0. The LHTES unit performance with PCM organic paraffin RT18 encapsulated in electrospun fiber matrices was analyzed. The study results show that longer internal fins shorten the melting and solidification time. Direct contact of PCM electrospun fiber matrix with 23 °C water showed instant melting, and the phase change process was accelerated by 99.97% in the discharging cycle.

**Keywords:** LHTES; PCM; numerical simulation; Comsol Multiphysics

## 1. Introduction

Heating, ventilation, and air-conditioning (HVAC) systems are designed to maintain a satisfactory indoor climate in residential and commercial buildings. Latent Heat Thermal Energy Storage (LHTES) with Phase Change Materials (PCM) is a promising technology for improving the efficiency of HVAC systems due to its high energy-storage density. When undergoing melting and solidification, PCM stores and releases a significant amount of latent heat (kJ/kg) at a relatively constant temperature. In this way, a relatively small volume of material facilitates the storage of a relatively large amount of energy within a narrow temperature range. Several studies [1–8] have reviewed TES applications with PCM. PCMs used in LHTES applications are selected based on the required thermal properties according to the application and climate conditions, e.g., the temperature range (°C) and the latent heat of fusion (kJ/kg).

Several finite element simulation studies [9–18] have addressed the influence of geometries on LHTES performance. The discharging thermal cycles of thermal energy storage with NaNO₃/KNO₃-PCM in an AISI 321 tube were studied by Zhang et al. [9]. Numerical analysis results and experimental data were in line, and inserts of metallic foam/sponge had an insignificant effect on the solidification rate of salt [9]. A high-temperature LHTES system for concentrated solar power (CSP) plants with magnesium chloride as PCM enhanced with graphite foam has also been analyzed by Zhao et al. [10]. The addition of

graphite foam in the PCM increased the exergy efficiency and improved the heat transfer processes [10]. Aadmi et al. [11] studied experimentally and numerically (with Comsol Multiphysics) a hot plate apparatus with PCM composites of epoxy resin paraffin wax. It was discovered that the container geometry affected the melting of PCM and a higher PCM content improved the LHTES capacity. KNO3–NaNO3 was encapsulated in a spherical shell, and no cracking was observed in the shell in an experimental and numerical investigation [12]. In another study by Arena et al. [13], the mushy zone of a finned double-pipe LHTES with paraffin RT35 was examined in three cases of heat transfer by convection and two cases for laminar and turbulent flow. Larger values for the mushy zone constant, representing the mushy area, led to a reduction of natural convection in the charging and discharging thermal cycles. A TES system with composite epoxy resin spherical shape paraffin wax RT27 was studied by Moulahi et al. [14], and the mushy zone proved to have an impact on the melting range. A numerical study of a PCM-air heat exchanger with dodecanoic acid was conducted by Herbinger et al. [15], and a higher heat transfer rate was observed for smaller heat exchanger channels and a higher air temperature. LHTES system performance under partial load in both charging and discharging cycles has also been studied by Arena et al. [16]. At a melting fraction of 0.75 and 0.90, the duration of the thermal cycles decreased up to 50% and the stored energy up to 30% [16]. The thermal properties of a PU-PCM composite were experimentally and numerically examined by Purohit and Sistla [17], and the study output indicated nucleation and crystallization domination of salt hydrate. Afsharpanah et al. [18–20] focused on several enhancement methods, such as porous foams, fins, and nanomaterials. A copper foam enhancement technique [18] increased the phase change rate by 92.5%.

In other studies [19,21–24], various thermal conductivity enhancement methods (e.g., examination of various geometries and addition of fins) have been analyzed for double-pipe LHTES systems. The effect of sinusoidal wavy fins has been studied by Shahsavar et al. [21,22] and achieved a melting/solidification time reduction of 43.49% and 17.81% with a wavy fin with amplitude and wavelength of 2 and 1 mm, respectively. The heat transfer enhancement with six configurations of different fin numbers and orientations of a double-pipe LHTES has been studied by Boulaktout [23], and it was shown that fewer fins with a proper orientation could be an effective geometry solution. The impact of fin type and orientation of a double pipe heat exchanger with N-eicosane as a PCM has been numerically analyzed by Nicholls et al. [24], and the transversal corrugated fin design exhibited shorter charging and discharging time. Anchor-type longitudinal fins [19] in shell and tube storage lead to a melting process slower up to 201.5% than the solidification process.

In experimental and analytical studies, PCM with a phase change temperature range of 15 to 20 °C has been reviewed in the author's previous work [8]. Several materials comprising the classes of salt hydrates, organic paraffins, organic fatty acids, and renewable-based oils in the phase change temperature range of 15 to 20 °C have been experimentally identified for their thermal properties in the author's previous studies [25–27]. Among the examined PCM, organic paraffin RT18 exhibited the most stable performance and the highest latent heat (J/g) after being subjected to 200 thermal cycles, equivalent to a six-month lifetime. PCM in emulsion, polymer, and electrospun fiber matrix have been analyzed experimentally [25–27]. The thermal properties of the electrospun fiber matrix of organic paraffin RT18 have also been reported [25].

Several studies have focused on the finite element design of LHTES systems, while the addition of fins has also been analyzed [21–24]. The present study examines the potential use of organic paraffin RT18 in pure and fiber form as a PCM in an LHTES system with different geometry configurations in both charging and discharging thermal cycles. The LHTES system is characterized by a one-step heat exchange from outdoor air to the PCM and from the PCM to the water. The project's overall aim is to develop a heat exchanger for night cooling applications in office buildings.

## 2. Materials and Methods

### 2.1. LHTES System Design

The aim of this project was to exploit the latent heat of fusion in an LHTES by using materials with a melting point around room temperature, so energy reduction for cooling can be achieved. In the current study, the encapsulated PCM layer in the LHTES facilitates the absorption of rejected heat during occupied hours and the release of heat to the environment at night. In the charging phase that occurs during night hours, cold outdoor air cools down the PCM, which changes state from liquid to solid. In the discharging phase during occupied hours, the PCM absorbs heat from the return water flow changing state from solid to liquid. The charging (solidification) and discharging (melting) diagrams of LHTES are presented in Figure 1. In the discharging phase, water is circulated through the inner copper pipe with a constant mass flow rate of 0.038 kg/s in a turbulent flow. The HTF (water) temperature was set to 23 °C at the inlet of the LHTES unit for discharge. In the charging phase, solidification was achieved through the air to the tube wall and PCM heat transfer; thus, the water flow was neglected. Steady conditions for air temperature are set at 10 °C in the charging/solidification case. The cold night air is circulated through fans when lamellas are open during the night, and the solidification of PCM progresses. In that way, natural and forced convection are assumed in the charging phase, and the water pump stops during the night, not circulating the water. The current analysis focuses on the most significant parameters for the LHTES unit characterization, such as charge and discharge cycle duration and energy stored and released. Thus, eleven different charging and discharging processes were simulated for different geometries and LHTES unit configurations, as shown in Table 1. In this numerical study, a double-pipe LHTES unit was analyzed. The unit includes an inner copper pipe with water flowing through it, surrounded by a PCM layer and an external copper pipe. In the charging phase, heat exchange is achieved from outdoor air to PCM, and in the discharging phase, from water to PCM.

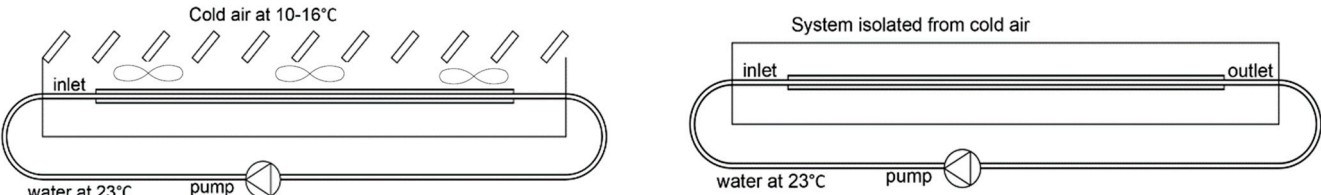

**Figure 1.** Charging (**left**) and discharging (**right**) diagram of LHTES.

**Table 1.** Geometry of models studied in cases 1–6.

| Case | Copper PCM Pipe Inner Diameter (m) | Thickness of PCM Pipe (m) | PCM Mass (kg) | Internal Fins | External Fins | Fins Width (m) | | Fins Height (m) | Distance between Fins (m) |
|---|---|---|---|---|---|---|---|---|---|
| 1 | 0.026797 m | 0.000889 m | 0.158 | | | - | | - | - |
| 2 | 0.0327914 m | 0.0010668 m | 0.257 | | | - | | - | - |
| 3 | 0.0387858 m | 0.0012446 m | 0.376 | | | - | | - | - |
| 4 | 0.0387858 m | 0.0012446 m | 0.362 | x | | 0.0065 m | | 0.0004 m | 0.0021 m |
| 5 | 0.0387858 m | 0.0012446 m | 0.352 | x | | 0.00975 m | | 0.0004 m | 0.0021 m |
| 6 | 0.0387858 m | 0.0012446 m | 0.364 | x | | 0.00975 m | | 0.0004 m | 0.0046 m |
| 7 | 0.0387858 m | 0.0012446 m | 0.376 | | x | 0.0065 m | | 0.0004 m | 0.0021 m |
| 8 | 0.0387858 m | 0.0012446 m | 0.376 | | x | 0.00975 m | | 0.0004 m | 0.0021 m |
| 9 | 0.0387858 m | 0.0012446 m | 0.376 | | x | 0.00975 m | | 0.0004 m | 0.0046 m |
| 10 | 0.0387858 m | 0.0012446 m | 0.352 | x | x | Int. fins | Ext. fins | 0.0004 m | 0.0021 m |
| | | | | | | 0.00975 m | 0.0065 m | | |
| 11 | 0.0387858 m | 0.0012446 m | 0.198 | | | - | | - | - |

The double-pipe LHTES unit configuration with and without fins analyzed in this study are presented in Figures 2–4. A parametric analysis was conducted for the system's geometry. The studied geometry of the model replicated a tubular double-pipe LHTES. Type M copper tubes with a length of 0.4 m were used for both pipes. The water pipe's

inner diameter (m) is 11.43 mm with a thickness of 0.635 mm. The double-pipe model's geometry is presented in Table 1. A novel LHTES unit consisting of a water pipe with encapsulated PCM electrospun fiber matrix has also been studied (Figure 5).

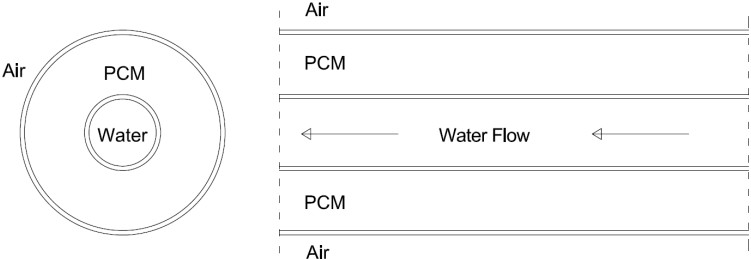

**Figure 2.** Configuration of double-pipe LHTES unit without fins.

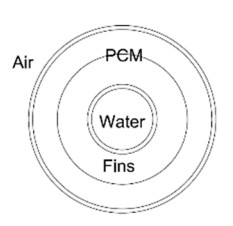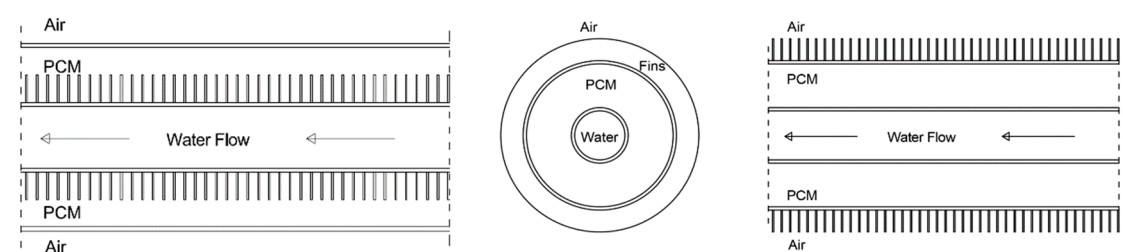

**Figure 3.** Configuration of double-pipe LHTES unit with fins on the internal pipe (**left**), LHTES unit with fins on the external pipe (**right**).

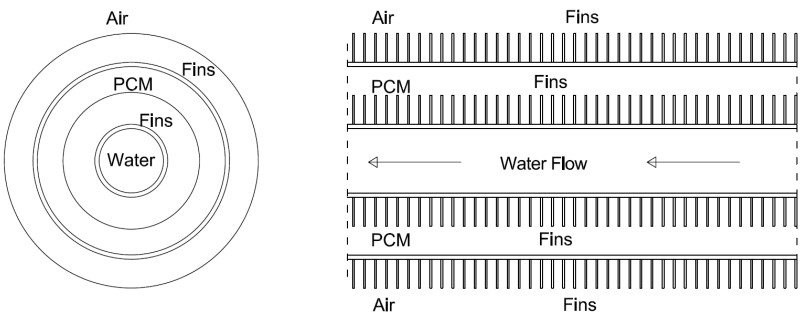

**Figure 4.** Configuration of double-pipe LHTES unit with fins on the internal and external pipes of the LHTES unit.

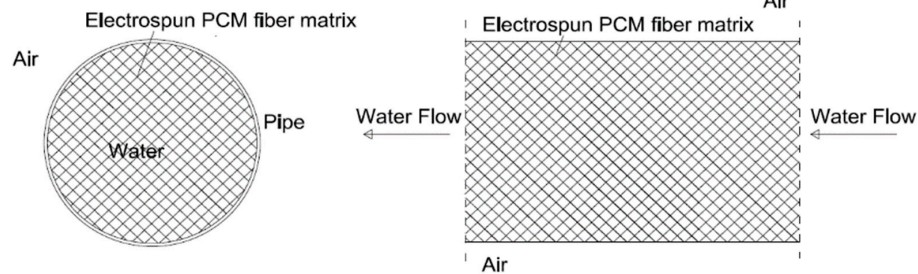

**Figure 5.** Configuration of the LHTES unit with RT18 electrospun fiber matrix as PCM.

The thermophysical properties provided by the manufacturers and the experimental thermal properties of organic paraffin RT18 in pure and electrospun fiber form are presented in Table 2. The PCM experimental properties have been previously analyzed [25,26], and the results were acquired in the laboratory of materials and production at Aalborg University. In the discharging phase, water was considered the heat transfer fluid (HTF), flowing with a turbulent flow and mass flow rate of 0.038 kg/s. The inlet water temperature for both charge and discharge was set to 23 °C. During the charging process, forced convection is

utilized with a convective heat flux of a heat transfer coefficient of 100 W/m²K and air temperature of 10 °C (the fan is blowing air over the LHTES). The LHTES system was isolated from air during discharging. Type M Copper pipes were selected for both unit's pipes with a better heat transfer rate than aluminum pipes.

**Table 2.** Thermophysical properties of organic paraffin RT18.

| Theoretical Properties | |
| --- | --- |
| Density (Solid) | 0.88 kg/L |
| Density (Liquid) | 0.77 kg/L |
| Heat Capacity (Solid) | 2 kJ/kg·K |
| Heat Capacity (Liquid) | 2 kJ/kg·K |
| Latent Heat of Fusion | 260 kJ/kg |
| Thermal Conductivity | 0.2 W/m·K |
| Melting Point | 18 °C |
| **Experimental Properties (Pure RT18)** | |
| Melting temperature | 17.5 °C |
| Solidification temperature | 15.4 °C |
| Latent heat of melting | 137.8 kJ/kg |
| Latent heat of solidification | 139.3 kJ/kg |
| **Experimental Properties (RT18 Electrospun Fiber Matrix)** | |
| Melting temperature | 17.3 °C |
| Solidification temperature | 15.2 °C |
| Latent heat of melting | 102.1 kJ/kg |
| Latent heat of solidification | 82.2 kJ/kg |
| Porosity of fiber matrix | 0.474 |

### 2.2. Numerical Model

The model was developed with Comsol Multiphysics 6.0 using different geometrical configurations in a two-dimensional axisymmetric interface. The finite element method and linear shape functions were used for all physics interfaces. The implicit backward differentiation formula (BDF) was the used stepping method with a time step of 10 s. The interval of $\Delta T_{1 \to 2}$ around the phase change temperature was investigated experimentally and set to 5K in melting and 2.5 K in solidification processes. Within the interval $\Delta T_{1 \to 2}$, there is a "mushy zone" with mixed material properties. Cases 1–3 use a double pipe unit with no fins, while cases 4–6 and 7–9 involve double pipe with internal and external fins, respectively. Case 10 addresses a double pipe with internal and external fins. Case 11 examines a single pipe encapsulating a PCM electrospun fiber matrix. An extra fine mesh was used in the Comsol model for a minimum error range, and the average element quality in all geometries was 0.81–0.89. In the discharging phase, we decided to couple together, in a time-dependent study, the two single-physics codes of turbulent flow and heat transfer in solids and liquids in a multiphysics system. This method transfers information between each module during the solution process. In the charging phase, a time-dependent study of heat transfer in solids and liquids was used. A k-ε model using the RANS (Reynolds-Averaged Navier–Stokes) equation was selected for the simulation of turbulent flow (Equation (1)). For the heat transfer in solids and liquids, the energy Equation (2) was used. The heat transfer process's effective density, heat capacity, mass fraction, and thermal conductivity in the phase change module are described by Equation (3)–(6).

$$\rho \frac{\partial u}{\partial t} + \rho (u \nabla) u = \nabla [-Pl + K] + F, \ \rho \nabla u = 0 \tag{1}$$

$$\rho C_P \frac{\partial T}{\partial t} + \rho C_P u \nabla_T + \nabla_q = Q, \quad q = -k \nabla_T \tag{2}$$

$$\rho = \theta_1 \rho_{ph1} + \theta_2 \rho_{ph2}, \quad \theta_1 + \theta_2 = 1, \tag{3}$$

$$C_P = \frac{1}{\rho} \left( \theta_1 \rho_{ph1} C_{P,1} + \theta_2 \rho_{ph2} C_{P,2} \right) + L_{1 \rightarrow 2} \frac{\partial a_m}{\partial T}, \tag{4}$$

$$a_m = \frac{1}{2} \frac{\theta_2 \rho_{ph2} - \theta_1 \rho_{ph1}}{\theta_1 \rho_{ph1} + \theta_2 \rho_{ph2}}, \tag{5}$$

$$k = \theta_1 k_1 + \theta_2 k_2 \tag{6}$$

## 3. Results and Discussion

### 3.1. Validation Model Analysis

Medrano et al. have studied different configurations of LHTES systems [28]. The parameters of two experiments with a turbulent flow of 0.38 m$^3$/h and 0.35 m$^3$/h were presented in Table 3. The double-pipe heat exchanger with encapsulated PCM in the annular space configuration previously presented [28] is the same configuration as the one analyzed in the current study. For this reason, the model created in Comsol Multiphysics was adjusted to have the same geometry and was validated with previous experimental data [24]. The parameters studied for the validation were the inlet and outlet temperature of the water, the temperature of the PCM-copper pipe wall, as well as the melting time of the PCM. The melting time of the PCM was calculated through the solid-to-liquid phase indicator for average surface in the numerical simulation. In the experimental process, an uncertainty of 30 s for the time interval was calculated. The percentage difference between the experimental and calculated melting time for the validation case is shown in Table 4. The percentage difference is calculated according to Equation (7). The percentage differences for the melting time are below 5% and 8.3%, which is considered a positive outcome.

$$\% \, difference = \frac{|Experimental \, value - Numerical \, value|}{Experimental \, value} \tag{7}$$

**Table 3.** Parameters of experiments.

| Experiment | Process | Temperature Conditions | | Turbulent Flow | PCM Properties | | Air Temperature (°C) |
|---|---|---|---|---|---|---|---|
| | | Water Inlet (°C) | PCM at Start (°C) | Volumetric Flow Rate (m$^3$/h) | Latent Heat (J/g) | Phase Change Temp. (°C) | |
| 1 | Charge (Melting) | 50 | 24 | 0.38 | 157 | 35 | 22–24 |
| 2 | Charge (Melting) | 50 | 24 | 0.35 | 157 | 35 | 22–24 |

**Table 4.** Experimental/Numerical melting time.

| Experiment | Mass Flow Rate (kg/s) | Experimental Melting Time | Calculated Melting Time | Percentage Difference |
|---|---|---|---|---|
| 1 | 0.1064 | 25,200 s | 26,470 s | 5% < 10% |
| 2 | 0.0980 | 24,060 s | 26,060 s | 8.3% < 10% |

In Figure 6a,b, the temperature difference between inlet and outlet water temperatures, as well the % differences of ΔT for the two mass flow rates of 0.1064 kg/s and 0.098 kg/s in the experiment and the simulation, are presented. ΔT water temperature difference % for different mass flow rates was calculated with Equation (7). In the experimental measurement, the inlet temperature at the beginning of the testing process fluctuates. As can be observed in Figure 6, the temperature difference between the inlet and outlet water

temperature graphs did not converge at the start of the experimental/simulation time. The temperature in the PCM-copper wall boundary and the temperature difference ($\Delta T$) between the experiment and the simulation for the two mass flow rates of 0.1064 kg/s and 0.098 kg/s are depicted in Figures 7a and 7b, respectively. The numerical simulation output is in good agreement with the experimental data. An error of 0.1 K was estimated for the calibrated temperature sensors in the experimental analysis.

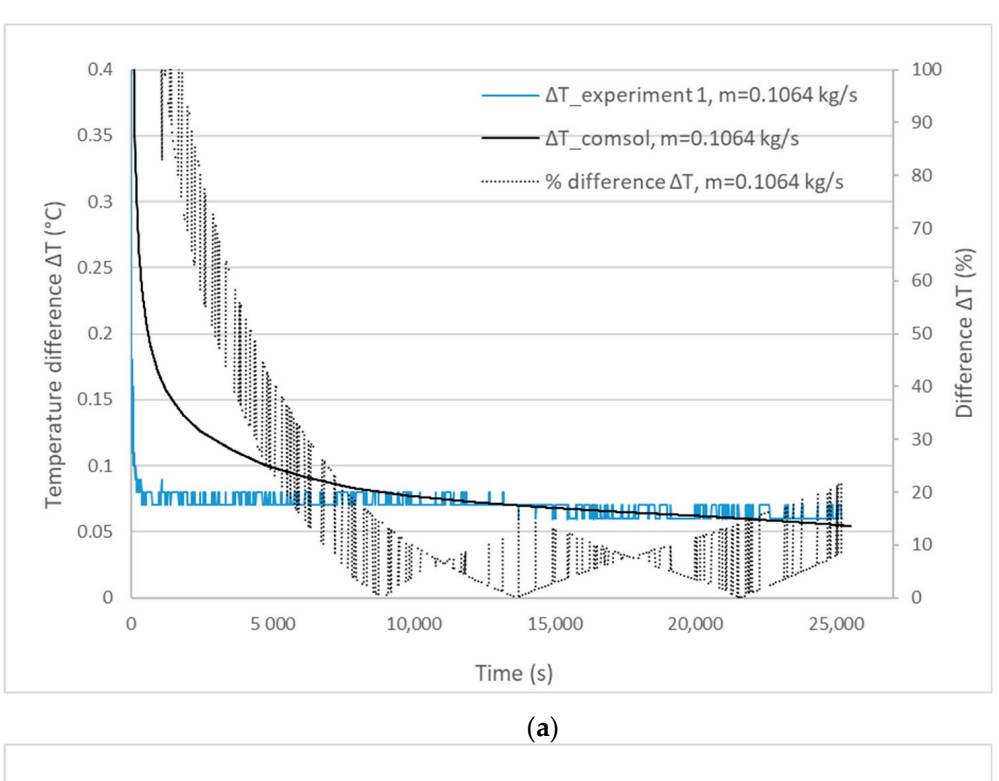

(**a**)

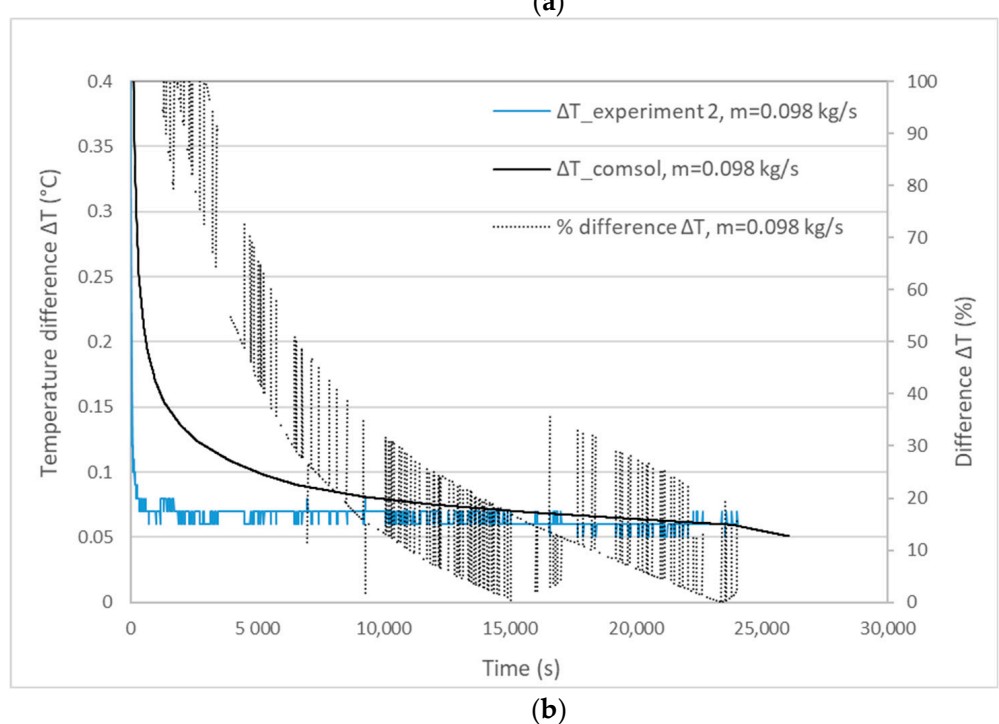

(**b**)

**Figure 6.** Inlet and outlet water temperature in experiments/numerical simulation for mass flow rate (**a**) m = 0.1064 kg/s, (**b**) m = 0.098 kg/s.

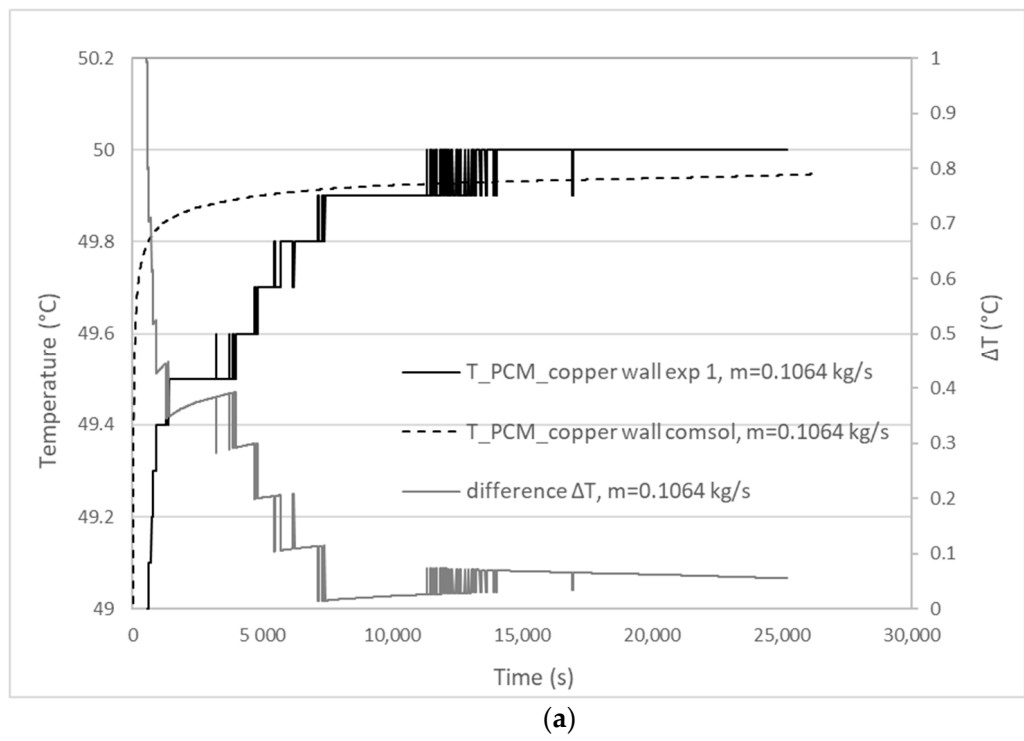

(**a**)

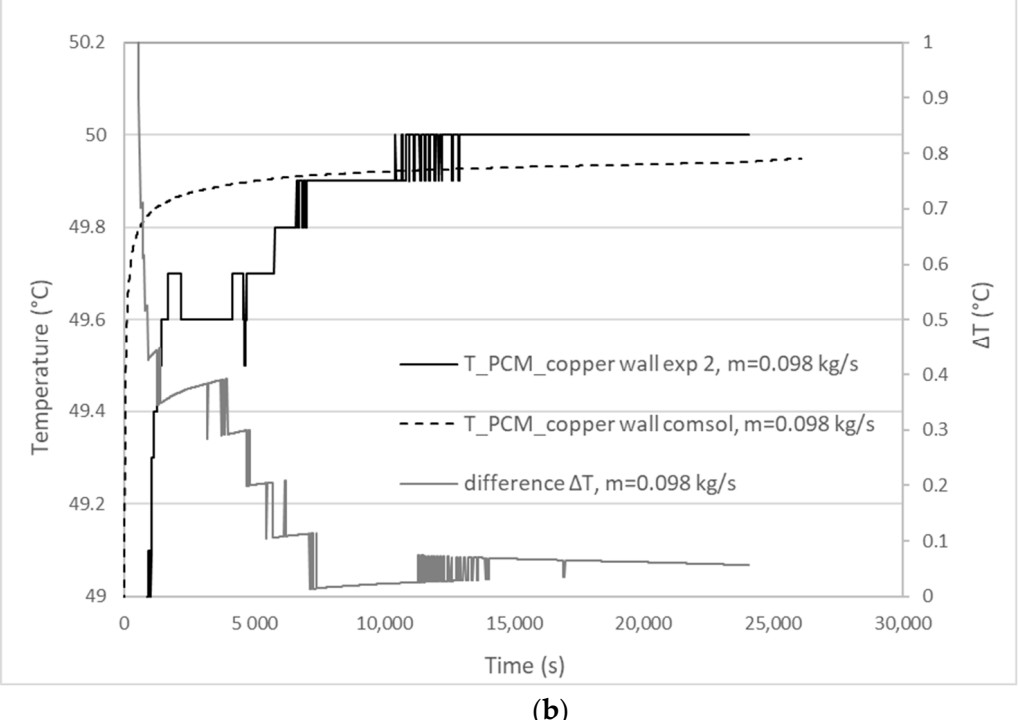

(**b**)

**Figure 7.** Temperature of PCM-copper wall in experiments/numerical simulation for mass flow rate (**a**) m = 0.1064 kg/s, (**b**) m = 0.098 kg/s.

### 3.2. Charging and Discharging Thermal Cycles

During the charging (solidification) process, the heat transfer mechanisms are conduction, forced convection (mechanical means-fans), and natural convection. In the discharging cycle (melting), conduction and natural convection occur in the phase change process. Natural convection plays a significant role at the beginning of the solidification process and when the melting fraction is increased in the melting process. The numerical simulation of solidification (charging) and melting (discharging) time is presented in Table 5. In cases 1,

2, and 3 (without fins), the discharging process lasts longer than the charging process. As expected, the duration of the charging and discharging processes gets longer with a higher PCM volume in the system. The PCM mass (kg) is reduced in the internal fins cases 4, 5, and 6 compared to base case 3 by 3.7%, 6.4%, and 3.2%, respectively. Due to PCM mass reduction, the charging and discharging time (cases 4–6 with internal fins) is significantly decreased compared to case 3. Longer internal fins reduce the charging and discharging time, while the melting/solidification time is increased with a longer distance between fins. The volumes of PCM in the cases of external fins are equal to the volume of PCM in case 3. In cases 7, 8, and 9 (with external fins), longer external fins decrease the melting and solidification time. When the distance between fins gets longer, the solidification time decreases, but the melting time increases (compared to case 8). The PCM mass reduction for case 10 with internal and external fins is estimated at 6.4%. As anticipated, the addition of internal and external fins (case 10) shows a reduction in time for both charging and discharging. The PCM mass in case 11 of the fiber matrix is reduced by 47.3% compared to base case 3. In case 11, the solidification time is shorter compared to case 3. The last case examined (analyzing the electrospun fiber matrix) exhibits a concise melting time of 4 s, which is expected due to the direct contact of the fiber matrix with the encapsulated PCM and water at 23 °C.

**Table 5.** Numerical simulation melting and solidification time.

| Case | Charging (Solidification) Time | Discharging (Melting) Time |
|:---:|:---:|:---:|
| 1 | 3310 s | 3420 s |
| 2 | 4090 s | 7070 s |
| 3 | 6940 s | 13,620 s |
| 4 | 5500 s | 3220 s |
| 5 | 4060 s | 1230 s |
| 6 | 4500 s | 1530 s |
| 7 | 5830 s | 14,590 s |
| 8 | 5710 s | 14,070 s |
| 9 | 4520 s | 14,510 s |
| 10 | 3040 s | 1040 s |
| 11 | 4780 s | 4 s |

The phase evolution at MF (melting fraction) equal to 0.75, 0.5, and 0.25 in charging and discharging processes for the eleven different geometries tested are presented in Figure 8. The blue-colored region represents the solid fraction in the phase evolution, and the red region represents the liquid fraction.

In the melting phase (discharging), the inner water tube wall is at a higher temperature, and the melted PCM is heated and moving due to buoyancy currents.

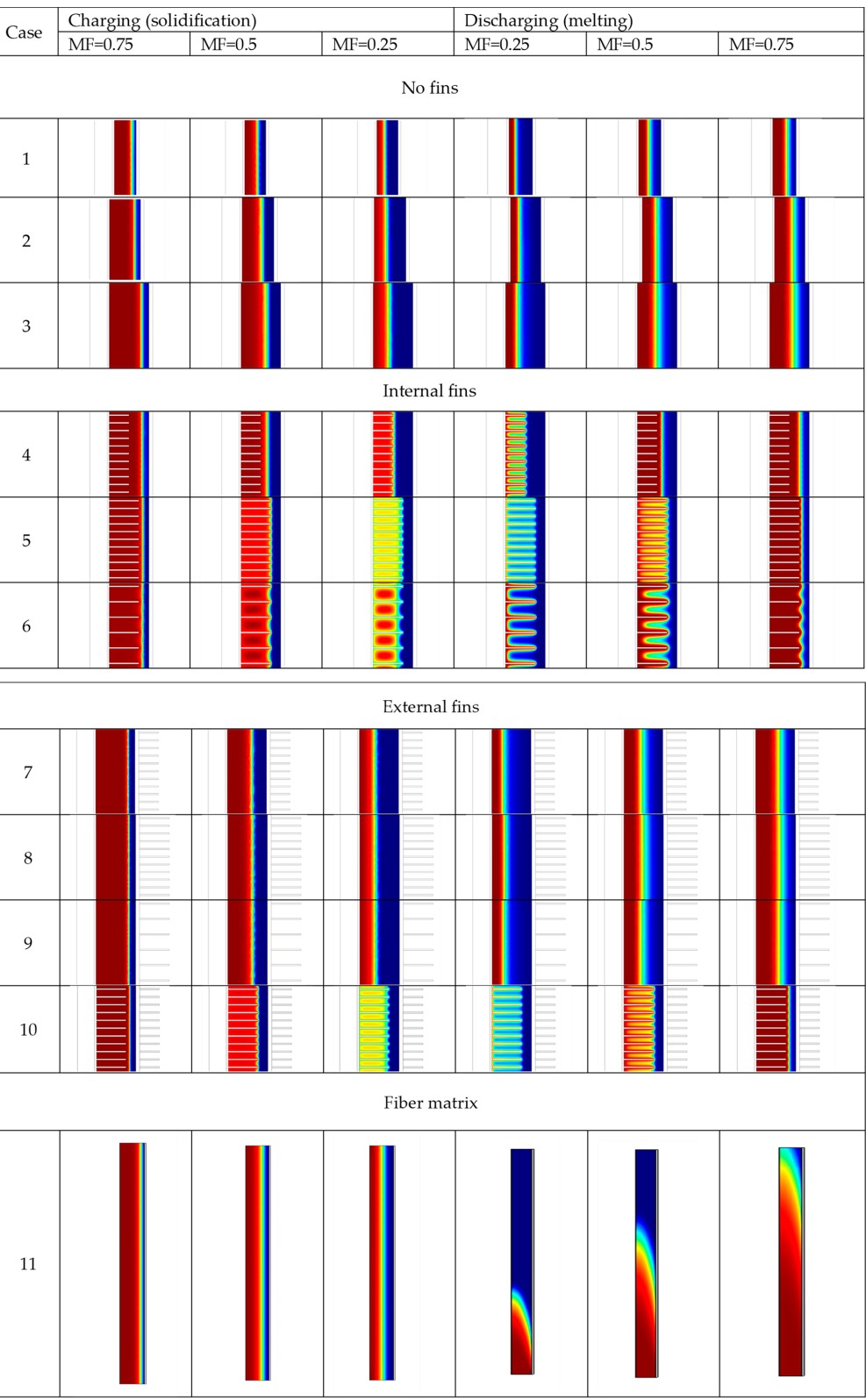

**Figure 8.** Phase evolution at MF = 0.75, 0.5 and 0.25.

The charge and discharge percentage during the thermal cycle duration are represented by the melting fraction (MF) graphs. The melting fraction is the ratio between the liquid/solid phase PCM volume and the total PCM volume. The MF graphs representing the variation between the solid-liquid phase of PCM and the melting front during the charging and discharging processes are depicted in Figure 9. The melting fraction is always 0 initially

for the discharging process (PCM is solid) and 1 for the charging process (PCM is liquid). In case 3, the MF curve (Figure 9a) represents a slower melting fraction evolution during the charging process than in all other cases. The external fins in cases 7, 8, and 9 MF curves (Figure 9c) converge in the charging and discharging process. In the discharging process, the MF evolution of internal fins in cases (4–6) (Figure 9b) is evidently faster than the MF evolution of the external fins in cases (7–9) (Figure 9c). Case 1 (Figure 9a) and case 10 (Figure 9d) follow the same trend during charging. Case 10 (Figure 9d), during the discharging, exhibits a faster MF evolution than case 1 (Figure 9a). Case 1 (Figure 9a) and case 11 (Figure 9e) converge during charging, while case 11 represents a swift MF evolution in the discharging phase.

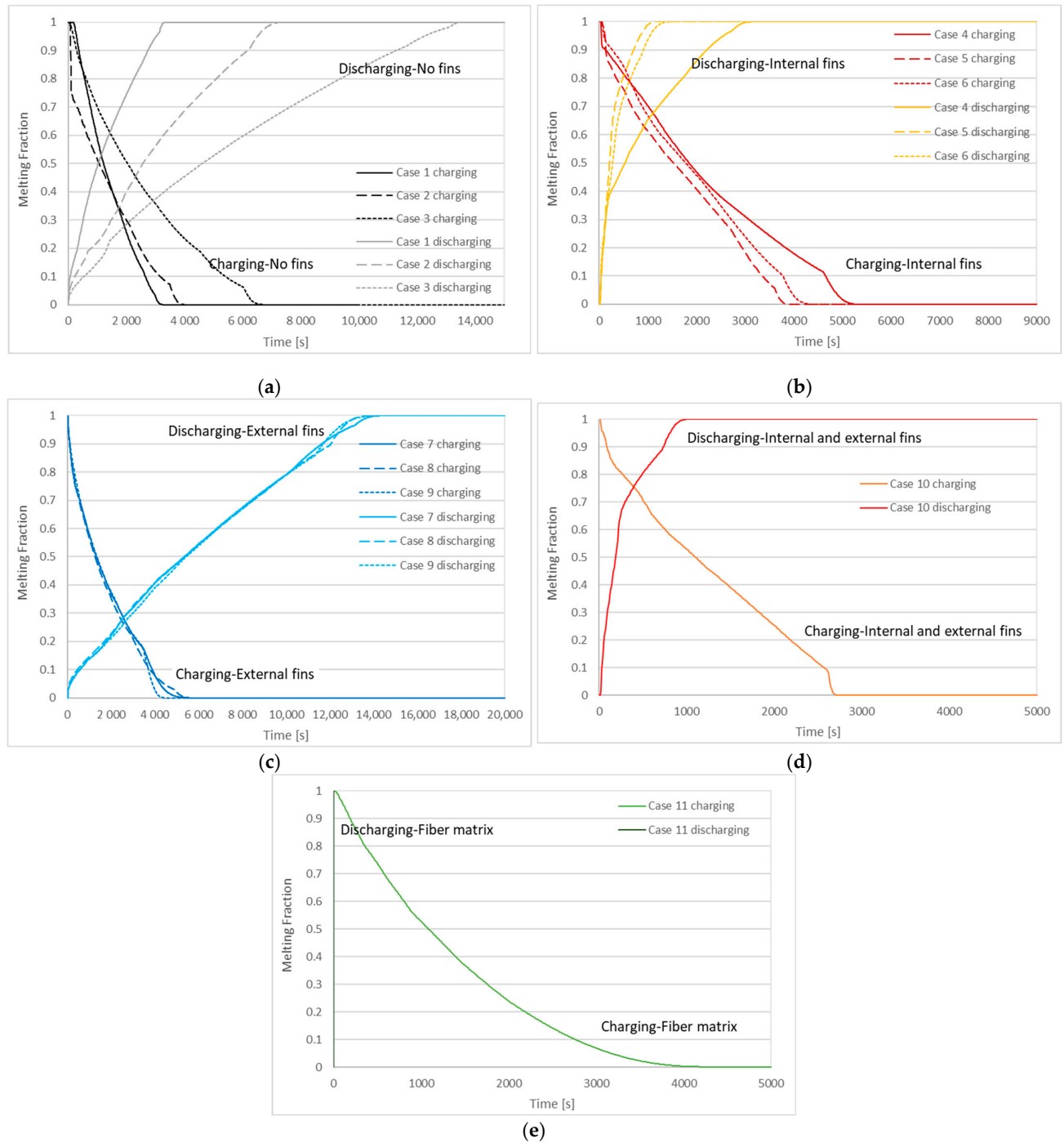

**Figure 9.** Melting fraction during charging and discharging processes (**a**) cases 1–3, (**b**) cases 4–6, (**c**) cases 7–9, (**d**) case 10, (**e**) case 11.

The surface average PCM temperature profile during the charging and discharging processes are depicted in Figure 10. In the charging phase, the temperature difference between the cold air (10 °C) and PCM drives the heat transfer rate inside the storage system. The decrease (charging) and increase in temperature versus time are analyzed in Figure 9. The solidification process is initiated at 15.4 °C for cases 1–10 (Figure 10a–d) and at 15.21 °C for case 11 (Figure 10e). The melting process starts at 17.51 °C for cases 1–10 (Figure 10a–d) and 17.33 °C for case 11 (Figure 10e). Throughout the melting process (discharging), sensible heat from the inner water tube is initially absorbed by PCM in a solid state. Then the PCM temperature increases, and natural convection is initiated when the melting process progresses.

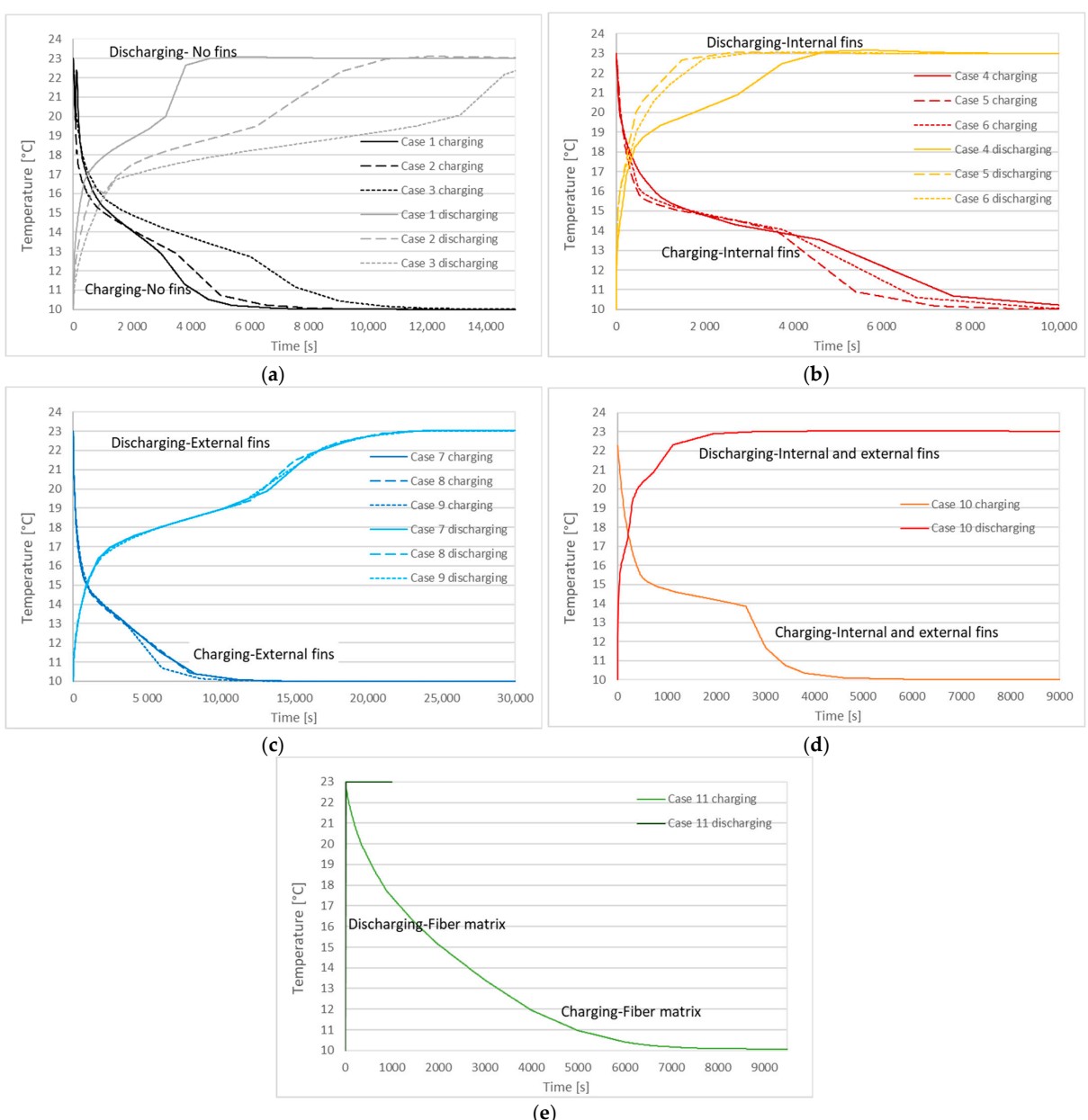

**Figure 10.** PCM temperature during charging and discharging processes (**a**) cases 1–3, (**b**) cases 4–6, (**c**) cases 7–9, (**d**) case 10, (**e**) case 11.

The evolution of total energy fluxes (W/m²) during the charging and discharging phases is depicted in Figure 11. The linear average energy flux for the charging (solidification) process for the right boundary PCM-wall-air is analyzed in Figure 11. For the discharging (melting) phase, the total energy flux (W/m²) of the left boundary PCM-wall-water is presented in

Figure 11. The heat flux flow (W/m²) evolves quickly in the simulation's beginning and then approaches zero. The simulated energy fluxes during charging and discharging display little variations. In the charging phase for cases 1–3 (Figure 11a), the smaller diameter LHTES (case 1) presented a lower heat flux and approached 0 more quickly at 3h. Additionally, in cases 5 and 6 with internal fins (Figure 11b), longer fins displayed lower energy flux and approached 0 at 3.3–3.6 h. the energy flux curves of cases 7–9 (Figure 11c) with external fins converge and present three times higher energy flux at the beginning of the simulation compared to other cases and reach 0 at around 4 h. Case 10 (Figure 11d) follows the same trend as cases 4–6. The energy flux of case 11 fiber matrix (Figure 11e) started at a higher value than cases with no fins and internal fins and reached 0 at 2.5 h. The discharging curves of energy flux cases 1–10 follow the same trend as the charging curves. However, the energy flux of case 11 with the electrospun fiber matrix approaches 0 at 10 s. This is explained due to direct-contact heat transfer of the water-electrospun fiber matrix. In both charging and discharging thermal cycles, external fins (cases 7–9) enhance the heat transfer of the LHTES system.

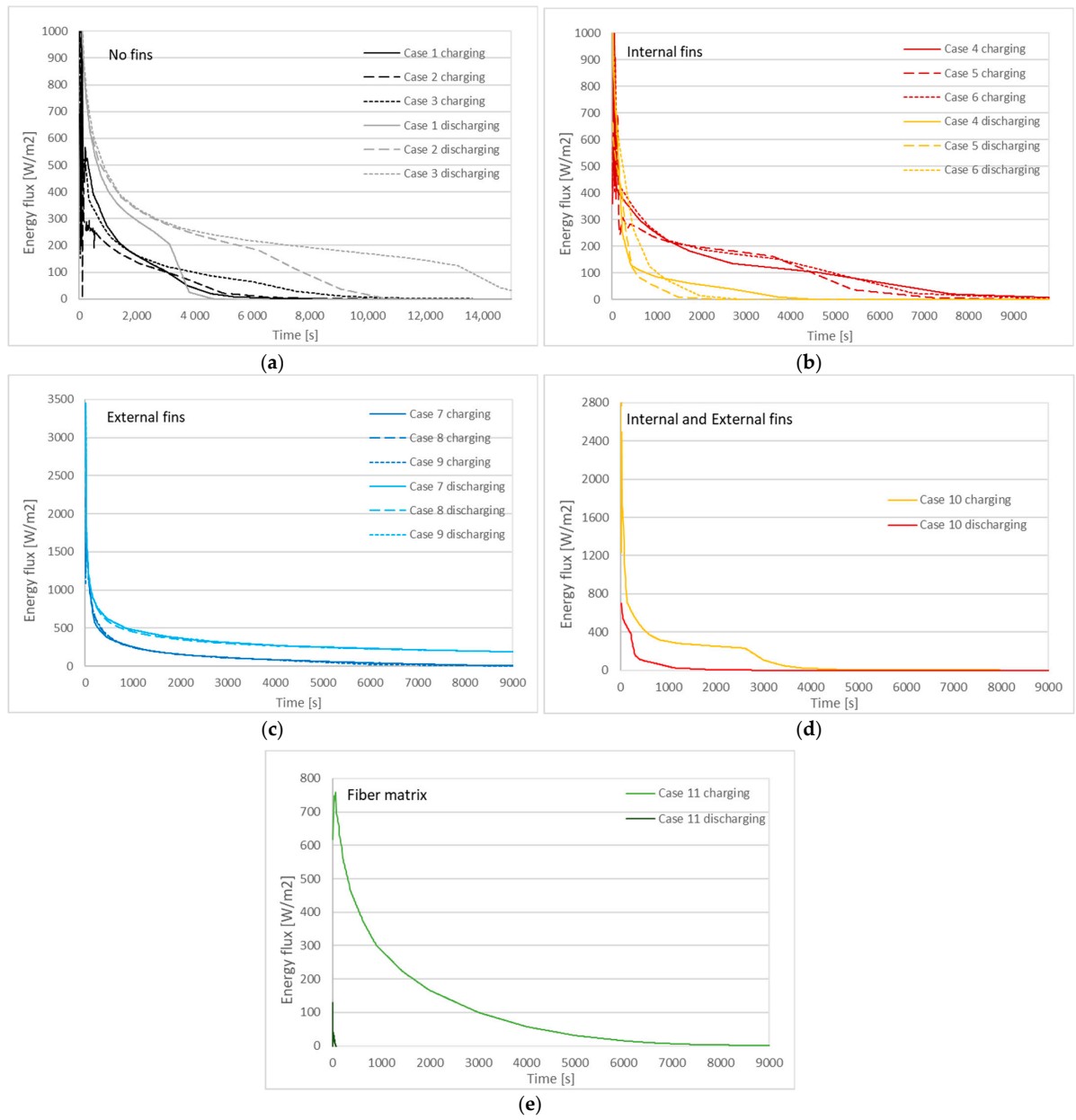

**Figure 11.** Energy flux (W/m2) during charging and discharging processes (**a**) cases 1–3, (**b**) cases 4–6, (**c**) cases 7–9, (**d**) case 10, (**e**) case 11.

The total enthalpy for the LHTES system during the charging and discharging processes are presented in Figure 12. The time required to store and release the energy (kJ/kg) corresponds to the solidification and melting time. As observed in Figure 12, the duration needed for storing and releasing the total latent heat is affected by the length and number of internal and external fins. More specifically, during charging and discharging in cases 1–3 with no fins (Figure 12a), the bigger the diameter of the PCM annular tube, the higher the stored enthalpy. In solidification (charging) in cases 4–6 with internal fins (Figure 12b), case 5 with longer fins exhibits lower total enthalpy than the other two cases. Total enthalpy curves of cases 7–9 (Figure 12c) follow the same trend as cases 4–6 during solidification. Throughout melting, the total enthalpy curves of cases 7, 8, and 9 converge, and case 9 presents the higher total enthalpy at the beginning of the discharging. In all cases with fins, the higher total enthalpy when the LHTES is charged is observed for the case with longer internal fins in a decreased number of fins. The total enthalpy of case 11 (Figure 12e) at the start of the charging phase and the end of the discharging phase is 64.6 kJ/kg.

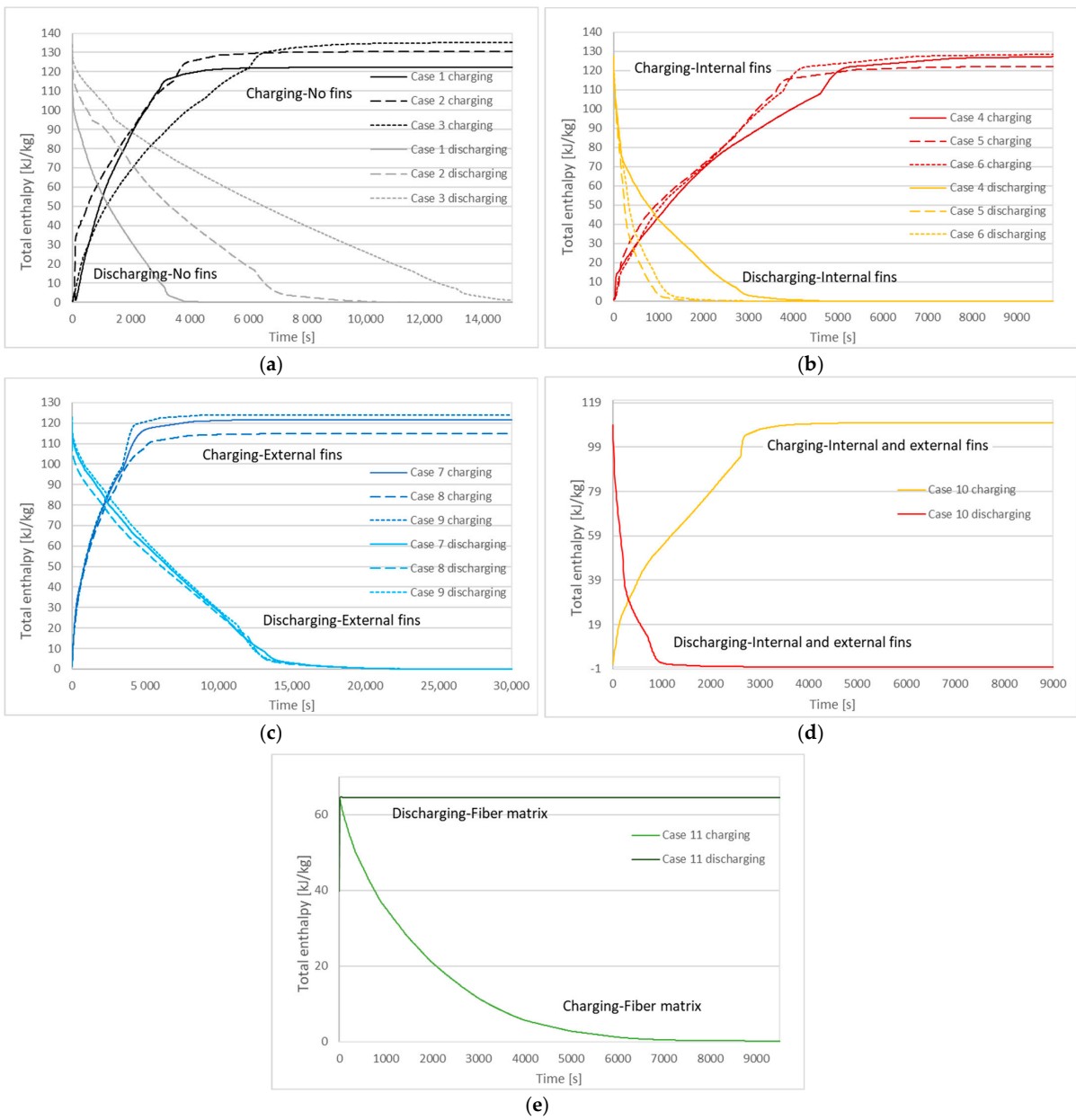

**Figure 12.** Total enthalpy of LHTES during charging and discharging processes (**a**) cases 1–3, (**b**) cases 4–6, (**c**) cases 7–9, (**d**) case 10, (**e**) case 11.

## 4. Conclusions

A numerical model of an LHTES unit was developed using Comsol Multiphysics 6.0, and a parametric analysis based on the geometry characteristics was conducted. The influence of external and internal fins was studied by examining ten cases of a double-tube heat exchanger and one single-tube LHTES. RT18, as a PCM, filled the annular space of the double tube, and its experimental melting/solidification latent heats (kJ/kg) and melting/solidification temperatures (°C) were used as input. RT18 electrospun fiber matrix, as a PCM, filled the LHTES tube in case 11. The impact of the number and length of fins on the LHTES system performance is observed using the results of liquid fraction, PCM temperature, energy flux, and total enthalpy. As the length of internal and external fins increases, the melting and solidification are accelerated. The highest energy fluxes (W/m$^2$) are displayed in cases with external fins. Case 1, with the lowest PCM volume, and case 10, with internal and external fins, exhibit the fastest solidification time of 0.92 h and 0.84 h, respectively. The numerical simulation's shortest discharging time is displayed for the electrospun fiber matrix case at 4 s due to the direct contact heat transfer of PCM matrix water. The phase change process was accelerated by 99.97% in the discharging cycle and by 31.12% in the charging cycle compared to the case with no fins of the same external tube diameter (case 3). A small-scale LHTES shall be constructed in future work, and an experimental evaluation should be conducted as additional research.

**Author Contributions:** Conceptualization, E.P. and P.F.; methodology, E.P. and A.A.; software, E.P.; validation, E.P., L.G. and P.F.; formal analysis, E.P.; investigation, E.P., P.F. and A.A.; resources, P.F.; data curation, P.F. and E.P.; writing—original draft preparation, E.P.; writing—review and editing, E.P., L.G., P.F., M.M., S.F., J.F. and A.A.; supervision, E.P., P.F. and A.A.; project administration, E.P.; funding acquisition, A.A. All authors have read and agreed to the published version of the manuscript.

**Funding:** The authors acknowledge the support provided by ELFORSK, a research and development program administrated by Danish Energy.

**Institutional Review Board Statement:** Not applicable.

**Informed Consent Statement:** Not applicable.

**Data Availability Statement:** Data are available upon request.

**Conflicts of Interest:** The authors declare no conflict of interest.

## Nomenclature

| Parameter | Description | Unit |
|---|---|---|
| ρ | Density | kg/m$^3$ |
| P | Pressure | Pa |
| u | Velocity | m/s |
| θ | Fraction/Indicator of phase transition | - |
| $C_P$ | Specific heat | J/kgK |
| $a_m$ | Mass fraction | - |
| k | Thermal conductivity | W/mK |
| HTF | Heat Transfer Fluid | - |
| LHTES | Latent Heat Thermal Energy Storage | - |
| PCM | Phase Change Material | - |

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
