# Peer review of "A Numerical Parametric Study of a Double-Pipe LHTES Unit with PCM Encapsulated in the Annular Space"

_sustainability, doi:10.3390/su142013317_

Round 1

Reviewer 1 Report

The authors proposed a numerical investigation for a double pipe LHTES unit 2 with PCM encapsulated in an annular space utilizing fins. The paper topic is interesting and the paper comprised new results. Before proceeding with the possible publication in the journal I recommend adding the following major revisions: 

1. The authors need to add the last author name to all of the cited papers in the Introduction part because they mentioned them just as numbers and this is uncommon.

2. In the Introduction part references [9]-[21] need to be rearranged by citing them from oldest to more recent instead of writing them irregularly.

3. What are the discretization methods used for the numerical solution in Comsol code?

4. There is a vague statement "Error! Reference source not found" that appears on pages 3,4,5,7,8,11,13, and 14? This maybe belongs to missing figures' captions and the authors need to fix all of them.

5. A description prior to each of equations (1)-(6) needs to be added. In addition, the terms on these equations need to be defined in a nomenclature.

6. The authors need to add relations to illustrate how the density and thermal conductivity of the PCM were estimated in terms of the melting fraction?

7. On page 11 Line 249 the authors need to add a mathematical relation for evaluating melting fraction via Comsol.

8. The authors need to check the utilization of tenses in their writing and check grammar for the whole paper.

Author Response

1.The authors need to add the last author name to all of the cited papers in the Introduction part because they mentioned them just as numbers and this is uncommon.

Response 1: Thank you for your comment. The first author names are now mentioned in the cited papers of the introduction section. Additionally, the references were adjusted.

  1. In the Introduction part references [9]-[21] need to be rearranged by citing them from oldest to more recent instead of writing them irregularly.

Response 2: The references [9]-[21] were rearranged.

  1. What are the discretization methods used for the numerical solution in Comsol code?

Response 3: The finite element method was used in the physics interface. In the finite element method, linear shape functions were used for both the turbulent flow and the heat transfer interfaces for cases 1-10 and for the flow through porous media and heat transfer interfaces of case 11.

  1. There is a vague statement "Error! Reference source not found" that appears on pages 3,4,5,7,8,11,13, and 14? This maybe belongs to missing figures' captions and the authors need to fix all of them.

Response 4: The missing figures and tables captions were fixed.

  1. A description prior to each of equations (1)-(6) needs to be added. In addition, the terms on these equations need to be defined in a nomenclature.

Response 5:  The description of equations (1)-(6) was added, and a nomenclature table including the terms was added.

  1. The authors need to add relations to illustrate how the density and thermal conductivity of the PCM were estimated in terms of the melting fraction?

Response 6: Thank you for your comment. The relations of density and thermal conductivity are defined by equations (3) and (6), respectively. The phase indicators for phases 1 and 2 are defined as pph1 and pph2.

  1. On page 11 Line 249 the authors need to add a mathematical relation for evaluating melting fraction via Comsol.

Response 7: The melting fraction is defined by  phase indicator 1 and phase indicator 2 and the equations defining the phase indicators are analyzed in the equations section.  

  1. The authors need to check the utilization of tenses in their writing and check grammar for the whole paper.

Response 8: The grammar was checked by the authors.

Reviewer 2 Report

The authors of this paper have presented an analysis of the melting and solidification processes in a triplex/double-tube heat exchanger through two-dimensional numerical investigations with a commercial CFD software package. They employed RT18 as the PCM and utilized the system with fins and fiber matrix as heat transfer enhancers. The numerical analysis was validated with experimental data. Overall, the paper looks interesting, it is well-written, and within the scope of the journal and this specific special issue, but some problems should be addressed by the authors before the paper is considered for publication.

1.       There are abundant research papers on the improvement of the phase change processes in double-pipe and triplex-tube heat exchangers with fins and porous media. What makes your research paper different from theirs? Please highlight the novelty in the abstract section and improve the discussions regarding the novelty of your paper in the last paragraph of the introduction.

2.       The authors have not specified the time-step size in their manuscript. This value should be given for all time-dependent research studies. Moreover, the authors should explain what was the termination criteria for each time step. For example, what residuals, how many iterations per time step.

3.       The authors are required to present two plots for the grid-size independence and time-step size independence analyses to show what was their basis for selecting the current meshing size and time-step size. Usually, these plots are required for both solidification and melting processes, but I believe presenting these plots just for the melting process would be enough (because of the more complex process due to the extreme buoyancy effects)

4.       Why didn’t the authors consider constant PCM volume for different cases? It is obvious that the addition of fins, reduces the volume of PCM, and therefore a smaller amount of PCM remains for the phase change. Such a comparison does not seem to be much reasonable, as the fins or even the fiber matrix might occupy a considerable volume within the storage tank. I understand that redesigning the cases and re-modeling all of them is extremely time-consuming, and I am not demanding such a major revision at this stage, but I expect the authors to at least specify the amount of PCM volume reduction due to the addition of fins or fiber matrix and recommend fixing the PCM volume as a future study at the conclusion section.

5.       In Lines 140-150, please add some discussions regarding the flow considered for air during the charging process. The authors have stated that the air is isolated during the discharging process, but they have not talked about the type of flow considered for air during the charging process. Is it stationary or moving? If the authors had decided to model the free airflow, how did they decide to what extent should they enlarge the domain that includes air?

6.       This is an optional comment, but the paper would be more fruitful if the authors could discuss how they intend to implement such a design in a building or a room. I can consider that horizontal tubes can be used on the floor or the roof but where do the authors intend to use vertical tubes?

7.       What mushy zone constant did the authors select for their study? The number should be given in the manuscript.

8.       The introduction section of the paper is poorly written. While there are numerous research papers published on similar topics, the literature review section has focused on papers that are mostly from before 2020, and there are many bulk references without specifically discussing the references. It is expected to use more up-to-date studies from 2022 in this section (DOIs: 10.1016/j.icheatmasstransfer.2022.106281; 10.1016/j.est.2022.104464; 10.1016/j.csite.2022.102421; 10.1002/er.7654)

9.       Lines 35 and 39-40: The citations should be merged. In other words, “[1]–[8]” should be replaced by “[1–8]”.

10.   Lines 39-40, consider changing the sentence “Although the geometry of the system is another significant parameter determining LHTES performance, only a few studies have addressed it and performed finite element simulations of LHTES systems.”, as there are at least hundreds of research papers discussing the influence of geometrical parameters on the performance of LHTES systems through finite element simulations.

11.   Although the paper is understandable, in some places it looks to be written carelessly. Please consider Lines 48-49: “…and the enhancement of PCM with metallic foam/sponge composites increased thermal conductivity both experimentally and numerically.”. The authors are urged to proofread the paper another time as some vague sentences can be easily rewritten.

12.   Line 55: “mush zone constant” => “mushy zone constant”.

13.   It is necessary to add some quantitative values to the abstract section of the study, preferably, using percentage instead of phase change time. For example, to what extent (what percentage), the phase change process was accelerated by adding the foam matrix? The improvement percentage by inserting the best type of fin can be also added to this section.

14.    The conclusion section is alright but it can be significantly improved. The quantitative results obtained by each of these improvement methods can be compared with the base case and some recommendations for future studies can be added at the end of the conclusion section.

By addressing these issues, the paper becomes much more suitable for publication in this journal.

Author Response

  1. 1. There are abundant research papers on the improvement of the phase change processes in double-pipe and triplex-tube heat exchangers with fins and porous media. What makes your research paper different from theirs? Please highlight the novelty in the abstract section and improve the discussions regarding the novelty of your paper in the last paragraph of the introduction.

Response 1: The purpose of this project is to demonstrate that the energy demand for cooling can be reduced compared to traditional system by using outdoor air and thermal energy storage with PCM. The present study examines the potential use of organic paraffin RT18 in pure and fiber form as a PCM in a LHTES system with different geometry configurations in both charging and discharging thermal cycles. A novel heat exchanger was designed and constructed which encapsulated a PCM layer to absorb the rejected heat from the building during occupied hours and release it to the environment during the night. The system is characterized by one-step heat exchange from outdoor air to the PCM and from the PCM to the water.

  1. The authors have not specified the time-step size in their manuscript. This value should be given for all time-dependent research studies. Moreover, the authors should explain what was the termination criteria for each time step. For example, what residuals, how many iterations per time step.

Response 2: The implicit backward differentiation formula (BDF) was the used stepping method. The time step was 10 sec.

  1. The authors are required to present two plots for the grid-size independence and time-step size independence analyses to show what was their basis for selecting the current meshing size and time-step size. Usually, these plots are required for both solidification and melting processes, but I believe presenting these plots just for the melting process would be enough (because of the more complex process due to the extreme buoyancy effects)

Response 3: The time step size was selected at 10 sec and the mesh size was set to an element size of extra fine in order to have an average element quality in all geometries is 0.81-0.89. Manual simulations by changing the mesh quality were performed and three combinations are presented below for the phase indicator versus time graph.

Extra fine mesh, time step 10sec

Normal mesh, time step 10sec

Extra coarse mesh, time step 10 sec

  1. Why didn’t the authors consider constant PCM volume for different cases? It is obvious that the addition of fins, reduces the volume of PCM, and therefore a smaller amount of PCM remains for the phase change. Such a comparison does not seem to be much reasonable, as the fins or even the fiber matrix might occupy a considerable volume within the storage tank. I understand that redesigning the cases and re-modeling all of them is extremely time-consuming, and I am not demanding such a major revision at this stage, but I expect the authors to at least specify the amount of PCM volume reduction due to the addition of fins or fiber matrix and recommend fixing the PCM volume as a future study at the conclusion section.

Response 4: In the current study,  cases 3-11 have the same inner and outer pipe diameters. Therefore a slightly different PCM volume for the cases of internal fins (case 4-6) and the case of internal and external fins (case 10). However, the mass of PCM is almost half compared to base case 3. The volumes of PCM in the cases of external fins are equal to the volume of PCM in case 3. Reduction of PCM mass (kg) in the internal fins cases 4,5,6 compared to base case 3 are 3.7%, 6.4% and 3.2% respectively. The mass reduction for case 10 with internal and external fins is estimated at 6.4%. Finally, the PCM mass in case 11 of the fiber matrix is reduced by 47.3% compared to base case 3.

  1. In Lines 140-150, please add some discussions regarding the flow considered for air during the charging process. The authors have stated that the air is isolated during the discharging process, but they have not talked about the type of flow considered for air during the charging process. Is it stationary or moving? If the authors had decided to model the free airflow, how did they decide to what extent should they enlarge the domain that includes air?

Response 5: During charging process forced convection is assumed with convective heat flux of a heat transfer coefficient of 100 W/m2K and air temperature of 10C (fan is blowing air over the LHTES).

  1. This is an optional comment, but the paper would be more fruitful if the authors could discuss how they intend to implement such a design in a building or a room. I can consider that horizontal tubes can be used on the floor or the roof but where do the authors intend to use vertical tubes?

Response 6: A small-scale LHTES shall be constructed, and an experimental evaluation should be conducted as additional research. The cooling system assembly consists of a fan, tubes, and a plywood casing. In future work an extended model of LHTES with horizontal tubes is planned to be installed in an office building and the long-term performance of the LHTES shall be evaluated.

  1. What mushy zone constant did the authors select for their study? The number should be given in the manuscript.

Response 7: The interval of   around the phase change temperature is investigated experimentally and set to 5K in melting and 2.5K in solidification processes. Within the interval of  , there is a “mushy zone” with mixed material properties and amush is calculated by a study controlled equation.

  1. The introduction section of the paper is poorly written. While there are numerous research papers published on similar topics, the literature review section has focused on papers that are mostly from before 2020, and there are many bulk references without specifically discussing the references. It is expected to use more up-to-date studies from 2022 in this section (DOIs: 10.1016/j.icheatmasstransfer.2022.106281; 10.1016/j.est.2022.104464; 10.1016/j.csite.2022.102421; 10.1002/er.7654)

Response 8:  Thank you for your recommendation. The recommended literature was taken into account.

  1. Lines 35 and 39-40: The citations should be merged. In other words, “[1]–[8]” should be replaced by “[1–8]”.

Response 9:  The citations are merged but the style required by the journal is [1]–[8].

  1. Lines 39-40, consider changing the sentence “Although the geometry of the system is another significant parameter determining LHTES performance, only a few studies have addressed it and performed finite element simulations of LHTES systems.”, as there are at least hundreds of research papers discussing the influence of geometrical parameters on the performance of LHTES systems through finite element simulations.

Response 10:  Thank you for your comment. We have changed the sentence.

  1. Although the paper is understandable, in some places it looks to be written carelessly. Please consider Lines 48-49: “…and the enhancement of PCM with metallic foam/sponge composites increased thermal conductivity both experimentally and numerically.”. The authors are urged to proofread the paper another time as some vague sentences can be easily rewritten.

Response 11:  Thank you for your comment. The paper was proofread, and vague sentences were rewritten.

  1. Line 55: “mush zone constant” => “mushy zone constant”.

Response 12:  Mush zone constant is replaced by mushy zone constant.

  1. It is necessary to add some quantitative values to the abstract section of the study, preferably, using percentage instead of phase change time. For example, to what extent (what percentage), the phase change process was accelerated by adding the foam matrix? The improvement percentage by inserting the best type of fin can be also added to this section.

Response 13:  The phase change process of the foam matrix (case 11) was accelerated by 13.09% in the charging and by 99.97% in the discharging process compared to the base case with no fins (case 3).

  1. The conclusion section is alright but it can be significantly improved. The quantitative results obtained by each of these improvement methods can be compared with the base case and some recommendations for future studies can be added at the end of the conclusion section.

Response 14:  Some quantitative results were added in the conclusion section and a recommendation for future studies was added in the end of the conclusion section.

Round 2

Reviewer 1 Report

In spite of the authors addressing most comments mentioned by the reviewer, and the paper quality is now improved, there are a few mistakes that need to be fixed prior to the publication. Hence, I recommend adding the following minor revisions:

1.      In the Introduction part, the authors need to erase all abbreviations that belong to the first author's name and keep just the last author name in references [9],[10],[13] ,[14],[15],[16],[17],[21],[22],[23],[24].

2.      On page 2, line 55 please remove the author name between the brackets (C. Parrado) and keep just the reference number [12].

3.      On the nomenclature, the authors defined the Pressure as rho and Density as p and this is uncommon so symbols for these parameters need to be reversed by defining p for pressure and rho for density. Also, please try to rewrite these parameters in Eqs. (1)-(3) and everywhere they used on the paper according to this note.

Author Response

  1. In the Introduction part, the authors need to erase all abbreviations that belong to the first author's name and keep just the last author name in references [9],[10],[13] ,[14],[15],[16],[17],[21],[22],[23],[24].

Response 1:  Thank you for your comment. All abbreviations that belongs to the first athor’s name were deleted.

  1. On page 2, line 55 please remove the author name between the brackets (C. Parrado) and keep just the reference number [12].

Response 2: The author name in the brackets was removed.

  1. On the nomenclature, the authors defined the Pressure as rho and Density as p and this is uncommon so symbols for these parameters need to be reversed by defining p for pressure and rho for density. Also, please try to rewrite these parameters in Eqs. (1)-(3) and everywhere they used on the paper according to this note.

Response 3: Thank you for your comment. The symbol for density rho is the Greek letter ρ and it will be sustained because the equations e.g. Navier Stokes are met in the literature with density – ρ. However capital letter P was used for pressure in order to avoid misunderstanding.

Reviewer 2 Report

The authors have presented a satisfactory response to most of my comments and concerns, however, there are still some comments that the authors have missed or not properly addressed yet:

1.       In my second comment, I asked “the authors should explain what was the termination criteria for each time step. For example, what residuals, how many iterations per time step.” The authors have ignored this comment and have not presented a response to this part of my comment.

2.       In the response to my third comment, the authors have thoroughly discussed the reason for selecting this specific meshing size, but it is clear that selecting the time-step size of 10 s doesn’t have a rational basis. How did the authors decide to select 10 s as the time-step size? Why not 20 s or 5 s? Have the authors tested these values? Plots should be presented for different time-step sizes as well. Also, it is recommended to merge the three plots presented in the response sheet (for mesh sizes of Extra fine, Normal, and Extra coarse) and include them in the manuscript for the readership. The readers should be informed why the authors have selected these sizes for the mesh network and time-step sizes. In other words, I believe meshing size and time-step size independence analyses are necessary parts of every time-dependent numerical investigation.

By addressing these remaining comments, the paper would be suitable for publication.

Author Response

  1. In my second comment, I asked “the authors should explain what was the termination criteria for each time step. For example, what residuals, how many iterations per time step.” The authors have ignored this comment and have not presented a response to this part of my comment.

Response 1: The termination criteria in comsol is defined by the Relative tolerance factor. In the current simulation study the tolerance factor was set to 0.005 was used as the maximum amount of error in the convergence criteria. The number of iterations per time step is 10.

  1. In the response to my third comment, the authors have thoroughly discussed the reason for selecting this specific meshing size, but it is clear that selecting the time-step size of 10 s doesn’t have a rational basis. How did the authors decide to select 10 s as the time-step size? Why not 20 s or 5 s? Have the authors tested these values? Plots should be presented for different time-step sizes as well. Also, it is recommended to merge the three plots presented in the response sheet (for mesh sizes of Extra fine, Normal, and Extra coarse) and include them in the manuscript for the readership. The readers should be informed why the authors have selected these sizes for the mesh network and time-step sizes. In other words, I believe meshing size and time-step size independence analyses are necessary parts of every time-dependent numerical investigation.

Response 2: Thank you for your comment. The time step size was selected at 10 sec, and the mesh size was set to an element size of extra fine to have an average element quality in all geometries is 0.81-0.89. Manual simulations by changing the mesh quality were performed, and three combinations were presented in the previous response to reviewer 2 for the phase indicator versus time graph. The three plots for mesh sizing were merged and are shown below in Figure 1. The same procedure was followed for the time step selection. Thus, Figure 2 presents a melting fraction graph for different time step selections of 5 sec, 10 sec, and 20 sec. It has been decided to upload these two graphs as supplementary material of the submitted manuscript. In this way  the graphs will be available for the readers.

Figure 1: Mesh sizing  with 10s  time step

Figure 2: Time step sizes for extra fine mesh

Round 3

Reviewer 2 Report

If have found the response given by the authors satisfactory and convincing, therefore, I recommend the publication of this manuscript in its current form.